# A molecular extraction process for vanadium based on tandem selective complexation and precipitation

Oluwatomiwa A. Osin[1], Shuo Lin[1], Benjamin S. Gelfand[1], Stephanie Ling Jie Lee[2,3], Sijie Lin [2,3] & George K. H. Shimizu [1]✉

Recycling vanadium from alternative sources is essential due to its expanding demand, depletion in natural sources, and environmental issues with terrestrial mining. Here, we present a complexation-precipitation method to selectively recover pentavalent vanadium ions, V(V), from complex metal ion mixtures, using an acid-stable metal binding agent, the cyclic imidedioxime, naphthalimidedioxime ($H_2CID^{III}$). $H_2CID^{III}$ showed high extraction capacity and fast binding towards V(V) with crystal structures showing a 1:1 M:L dimer, $[V_2(O)_3(C_{12}H_6N_3O_2)_2]^{2-}$, **1**, and 1:2 M:L non-oxido, $[V(C_{12}H_6N_3O_2)_2]^-$ complex, **2**. Complexation selectivity studies showed only **1** and **2** were anionic, allowing facile separation of the V(V) complexes by pH-controlled precipitation, removing the need for solid support. The tandem complexation-precipitation technique achieved high recovery selectivity for V(V) with a selectivity coefficient above $3 \times 10^5$ from synthetic mixed metal solutions and real oil sand tailings. Zebrafish toxicity assay confirmed the non-toxicity of **1** and **2**, highlighting $H_2CID^{III}$'s potential for practical and large-scale V(V) recovery.

Vanadium is a rare and refractory metal with desirable physicochemical properties, including high melting point, high hardness, and corrosion resistance[1,2]. Consequently, vanadium's importance extends across various industries, including manufacturing, construction, aerospace, and renewable energy[3,4]. Due to depleted primary sources, environmental concerns about terrestrial extraction, and higher demand, recycling vanadium from alternative sources, such as refining waste streams, is needed[4,5]. Vanadium ions—predominantly pentavalent vanadium, V(V)—from such waste streams can contaminate soil, groundwater, and potable water[6]. Hence, vanadium recovery from waste offers both economic and environmental benefits.

V(V) has been extracted from aqueous solutions using porous sorbents[7–9], ionic liquids[10], colloids[11,12], nanoparticles/nanocomposites[13–16], and modified composites[17,18], with impressive capacities ranging from 240 to 712.4 mg g$^{-1}$. However, a comprehensive

evaluation of material's selectivity for V(V) in the presence of competing metal ions (Fe(III), Cr(III), Cu(II), Ni(II), Zn(II), etc.)[19] is limited and could impact these capacities. Furthermore, given the acidic nature of mining waste streams containing V(V)[20], it becomes crucial to evaluate the effectiveness of materials under harsh acidic (typically pH <2) aqueous waste solutions to determine their practical applications and technological potential.

Conventional hydrometallurgical approaches (e.g.,: solvent extraction, ion-exchange, or adsorption) separate V(V)[21–23] but suffer from limitations, including low affinity and selectivity, high extractant dosage, limited recyclability and toxic effluent generation[24–27]. Therefore, developing new materials and viable methods for recovering V(V) from acidic wastes is crucial. Chelates, characterized by a minimum of two donor atoms, emerge as effective extractants for V(V) recovery from waste materials[28–31]. These compounds form stable complexes

[1]Department of Chemistry, University of Calgary, Calgary, AB T2N 1N4, Canada. [2]College of Environmental Science and Engineering, Biomedical Multidisciplinary Innovation Research Institute, Shanghai East Hospital, Tongji University, 1239 Siping Road, Shanghai 200092, China. [3]Key Laboratory of Yangtze River Water Environment, Shanghai Institute of Pollution Control and Ecological Security, Tongji University, 1239 Siping Road, Shanghai 200092, China. ✉e-mail: gshimizu@ucalgary.ca

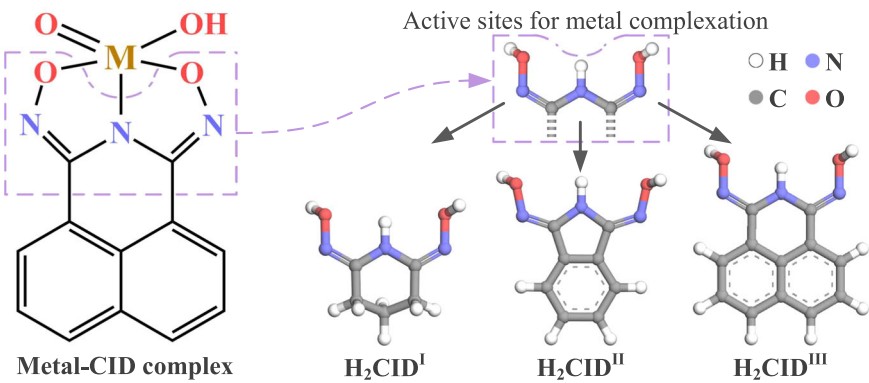

**Fig. 1 | Cyclic imidedioximes and metal binding mode.** Glutarimidedioxime (H$_2$CID$^I$), phthalimidedioxime (H$_2$CID$^{II}$), and naphthalimidedioxime (H$_2$CID$^{III}$).

with V(V) ions, augmenting the affinity of functional sites for the V(V) species. However, there is still a deficiency in the selectivity for V(V). Wołowicz and Hubicki's investigation showed that a macroporous weakly basic chelating ion exchange resin featuring a polyamine functionality demonstrated superior V(V) removal capabilities in both simulated and actual wastewater, outperforming other tested materials. Nevertheless, the authors addressed the chelator's affinity for Fe(III) ions throughout the extraction process[32]. Similar findings have been reported in the literature[28,33–36]. Consequently, advancing chelating agents specifically tailored for V(V) is necessary.

The Cyclic imidedioximes (CIDs) (Fig. 1 and Supplementary Fig. 1a) are metal chelating agents that have been actively studied for metal recovery from wastewaters. Recently, H$_2$CID$^I$ has been explored for removing uranium (U$^{IV}$ and U$^{VI}$)[37–39] cerium (Ce$^{IV}$)[40,41], plutonium (Pu$^{IV}$)[42], and neptunium (Np$^V$)[43] from nuclear wastewater. However, thermodynamic and spectroscopic studies revealed H$_2$CID$^I$ has higher affinity for V(V)[44,45]. Unfortunately, under acidic aqueous conditions typical for mineral processing, the CID moiety degrades (Supplementary Fig. 1b). Aromatic resonance stabilization has been observed to enhance the acid stability of the oxime and imide moieties, thus H$_2$CID$^{II}$ was synthesized[46]. However, the 5-membered ring backbone alters the metal-coordination bite angle and weakens binding with V(V). H$_2$CID$^{III}$ presents a noteworthy prospect, maintaining the ideal bite angle in relation to H$_2$CID$^I$, and containing two aryl rings, providing aromatic resonance stabilization, thereby emerging as the chelator of primary interest in this work.

Herein, we report a more scalable synthesis of an acid-stable, naphthalene-based CID material (H$_2$CID$^{III}$) and its use in an unsupported vanadium complexation-precipitation extraction technique from acidic solutions (Supplementary Fig. 2a–d)[47]. Additionally, we present two single crystal structures, a 1:1 M:L dimer, [V$_2$(O)$_3$(C$_{12}$H$_6$N$_3$O$_2$)$_2$]$^{2-}$, **1** and 1:2 M:L non-oxido [V(C$_{12}$H$_6$N$_3$O$_2$)$_2$]$^-$ complex, **2** (Fig. 2). The performance of H$_2$CID$^{III}$ for complexation and precipitation was investigated through batch experiments, encompassing pH-dependent effects, equilibrium, and kinetic studies. The mechanisms governing complexation, precipitation, and selectivity were studied using nuclear magnetic resonance (NMR) and Fourier transform infrared spectroscopy (FT-IR) alongside density functional theory (DFT) methods. These results corroborated the high acid stability of the V-CID$^{III}$ complexes. Here, V-CID$^{III}$ refers to pentavalent vanadium ion bound to the fully deprotonated ligand. V-CID$^{III}$ complexes are anionic, as opposed to competing metal ion complexes, enabling V(V) selective precipitation through pH adjustment, obviating the need for a solid support. Moreover, the selective complexation and precipitation procedures were successfully applied to recover V(V) from industrial oil sands tailings wastewater. To ensure the environmental compatibility of the extraction process, a zebrafish toxicity assay was conducted for **1** and **2**, confirming viability for practical applications in V(V) recovery.

## Results

### Synthesis and characterization

H$_2$CID$^{III}$ was previously reported[48] via a 3-step 4-day synthesis in 28% yield. For scalability, a 2-step 24-h synthesis was developed that gave an overall yield of 79% and eliminating chromatographic separation (see the "Methods" section, Supplementary Discussion 1 and Supplementary Fig. 3). The crystal structure of H$_2$CID$^{III}$ and spectroscopic analyses (Supplementary Figs. 4–7 and Supplementary Table 1) confirmed the pure synthesis of the ligand. As imputed by the aromatic resonance stabilization, H$_2$CID$^{III}$ demonstrated high resistance to acid-catalyzed hydrolysis at room temperature, suggesting promise for applications in metal recovery (Supplementary Discussion 2 and Supplementary Fig. 8).

Two distinct crystals of V(V) complexes were synthesized by reacting stoichiometric amounts of sodium metavanadate (NaVO$_3$) and H$_2$CID$^{III}$. The structures of these two crystals have been resolved through single crystal X-ray diffraction (SC-XRD) and Rietveld refinement (Fig. 2 and Supplementary Tables 2 and 3 and XRD patterns shown in Supplementary Fig. 9). The 1:1 V-CID$^{III}$ complex, Na$_2$[V$_2$(O)$_3$(C$_{12}$H$_6$N$_3$O$_2$)$_2$], Na$_2$**1** (Fig. 2a) and 1:2 V-CID$^{III}$ complex, Na[V(C$_{12}$H$_6$N$_3$O$_2$)$_2$], Na$_1$**2** (Fig. 2b) were respectively obtained by reacting stoichiometric amounts of NaVO$_3$ and H$_2$CID$^{III}$ in an H$_2$O/MeOH solution, followed by freeze-drying, redissolution in acetone, and crystal formation via ether vapor diffusion (see the "Methods" section). **1** is a one-dimensional chain of V(V) dimers bridged by sodium ions (Fig. 2c) and exhibits a distorted square pyramidal geometry with two V(V) centers sharing a μ-oxo group (Fig. 2a). Each V(V) center is coordinated to a triply deprotonated CID$^{III}$, via the imide N atom and the two oxime O atoms. The aromatic moieties π-stack with an inter-ring distance of 3.7 Å, forming the dimer. **2** exhibits a distorted octahedral structure with two fully deprotonated CID$^{III}$ ligands coordinated to a bare V(V), forming a non-oxido complex (Fig. 2b). Each complex in the extended structure is orthogonal to the next, creating a herring-bone arrangement of naphthalene rings. The complexes are bridged by a sodium ion, forming a one-dimensional chain (Fig. 2d). Notably, complete deprotonation of the imide (−C−N(**H**)−C−) and both oxime groups (−C=N−O**H**) favors strong bonding and the observed formation of the anionic non-oxido V-CID$^{III}$ complex. In contrast, Fe-CID$^{III}$ complex, **3**, contains deprotonated imide and protonated oximes, giving a neutral Fe(CID$^{III}$)(Cl)$_2$ complex (Supplementary Fig. 10 and Supplementary Table 4). The spectroscopic investigation of the V-CID$^{III}$ complexes (Supplementary Discussion 3 and Supplementary Figs. 11 and 12) formed the basis for the extraction process of V(V) developed in this study.

### Pristine vanadium complexation and precipitation studies

The performance of H$_2$CID$^{III}$ (Fig. 3a) for V(V) recovery was investigated by varying pH, metal concentration, and time. The pH-dependent V(V)

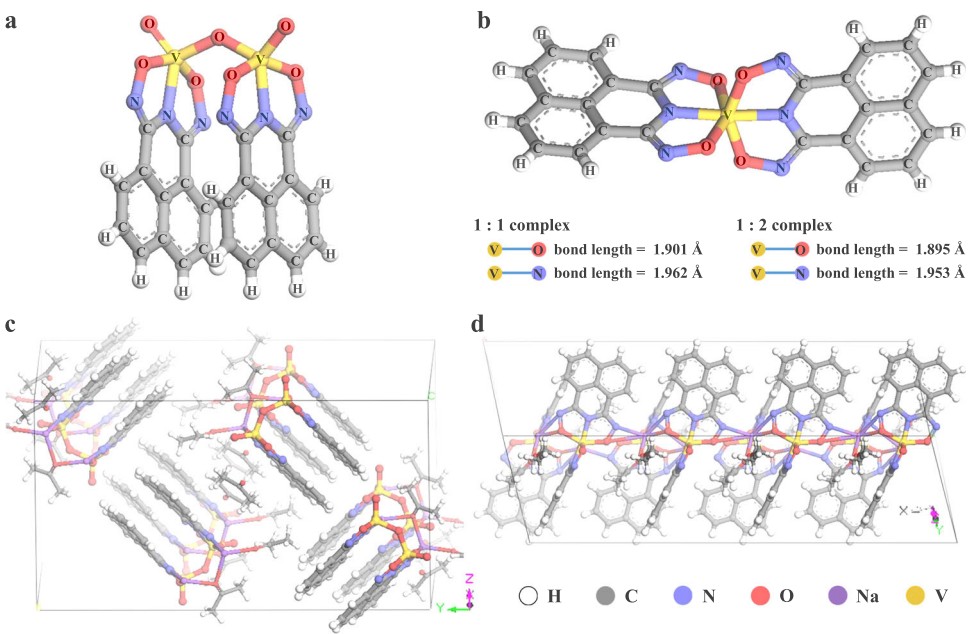

**Fig. 2 | Crystal structures of V-CID^III complexes.** Asymmetric units. **a** 1:1 V-CID^III complex, **1**. **b** 1:2 V-CID^III complex, **2**. Unit cell configurations. **c** 1:1 V-CID^III complex, **1**. **d** 1:2 V-CID^III complex, **2**. Countercations omitted for clarity. (Inset: bond lengths between the metal ion and O and N donor atoms).

extraction was evaluated via inductively coupled plasma optical emission spectrometer (ICP-OES) analyses to determine the V(V) concentration in the pre- and post-extraction solutions (see the "Methods" section). The acid stability and anionic nature of the V-CID^III complexes (Fig. 3b) enable facile and selective precipitation of V(V) through pH adjustment (Fig. 3c). $H_2CID^{III}$ exhibited high V(V) complexation from pH 5 to 12, with efficiency exceeding 99% and capacity reaching 80 mg g$^{-1}$ (Fig. 3d), underscoring the robust coordination between V(V) and $H_2CID^{III}$. Subsequently, the effect of pH on precipitation of V-CID^III was assessed while maintaining complexation pH at 8–9. The precipitation increased with acidity and achieved maximum efficiency (~99%) at and below pH 3 (Fig. 3d). These results reveal that V(V) can be bound and precipitated under acidic conditions using CID^III (inset in Fig. 3d).

The extraction capacity of $H_2CID^{III}$ for V(V) was determined in the complexation and precipitation steps, as explained previously, under pH 8 and 3, respectively. The extraction isotherm data fits with the Langmuir model (Fig. 3e and Supplementary Table 5), yielding a coefficient of determination ($R^2$) above 0.95. The maximum extraction capacity ($q_m$) of $H_2CID^{III}$ for V(V) reached 205.4 mg g$^{-1}$, surpassing that of many common V(V) adsorbents (Supplementary Table 6). Additionally, the Langmuir constant ($b$) value of 2.71 l mg$^{-1}$ indicated a high affinity of $H_2CID^{III}$ for V(V) compared to a wide range of V(V) adsorbents (Supplementary Table 6). During the sorption experiments, the metal to ligand stoichiometry resulted in two complexes, **1** and **2**, with corresponding equilibrium extraction amounts ($q_e$) values of 200 mg g$^{-1}$ and 80 mg g$^{-1}$. The inset of Fig. 3e displays the $^1H$ NMR spectra of pristine $H_2CID^{III}$, **1** and **2**. All samples exhibited three $^1H$ signals, comprising two doublets (8.06–8.16 ppm) and a triplet (7.65–7.62 ppm), assigned to the aromatic protons of the ligand. This is consistent with the spectra obtained from the crystals (Supplementary Fig. 11). A detailed discussion on kinetics, regeneration and reusability is provided in Supplementary Discussion 4 and 5 and Supplementary Figs. 13–16.

## Vanadium selectivity studies

Capacity, kinetics, and selectivity are interconnected properties of sorptive separations. For V(V), selectivity is paramount since waste streams contain a highly complex mixture of metal ions. An initial control experiment was undertaken to evaluate the complexation-precipitation method for various single-metal ion solutions (Fig. 4a) and mixed-metal ion solutions (Supplementary Fig. 17). In both types of experiments, only complexation solutions containing V(V) exhibited precipitation following acidification. Quantitative evaluation of selectivity was conducted by varying pH and time in a synthetic acidic solution containing V(V), Fe(III), Cr(III), Cu(II), Ni(II), and Zn(II). Leveraging the high affinity of $H_2CID^{III}$ for V(V), a one-step extraction approach was adopted to optimize the economic feasibility of the process. This method occurred under acidic conditions (pH 1.5) and involved the simultaneous occurrence of the complexation and precipitation steps. Extraction isotherm and kinetic experiments were performed to assess the effectiveness of $H_2CID^{III}$ under these specified conditions (refer to Supplementary Figs. 18 and 19). These results establish the practicality and viability of a one-step extraction V(V) process with $H_2CID^{III}$.

With an initial equimolar concentration of 1 mg l$^{-1}$ for each metal ion, $H_2CID^{III}$ (0.03 g, 0.15 g l$^{-1}$) displayed impressive selectivity, exhibiting a high V(V) adsorption capacity of 7 mg g$^{-1}$ within 30 min at pH 1.5, while other metals were scarcely recovered (Supplementary Figs. 20 and 21). The selectivity coefficient for V(V) versus Fe(III), Cr(III), Cu(II), Ni(II), and Zn(II) was high (Supplementary Table 7), highlighting the notable performance of $H_2CID^{III}$ compared to other V(V)-extractants (Supplementary Table 8). Considering the possibility of competing metal cations being at higher concentrations than V(V) in some waste sources, a selectivity assessment was conducted in the presence of 5-, 10-, and 20-fold excess of competing metal ions at pH at 1.5. The competitive binding assay revealed that, even with a 20-fold excess, Fe(III), Cr(III), Cu(II), Ni(II), and Zn(II), did not significantly compete with V(V) (Fig. 4b).

## Vanadium extraction mechanisms

To gain insight into the mechanisms of complexation, precipitation and selectivity, SC-XRD, potentiometric titration, Fourier transform infrared spectroscopy (FT-IR) and $^1H$ NMR spectroscopy were employed. The negative charge of complexes **1** and **2** in acidic solutions were confirmed by potentiometric titration beginning at solution pH above 10. At such high pH values, the major V(V) species present is $VO_3OH^{2-}$ (Supplementary Fig. 22a, b), which protonates during complexation with $H_2CID^{III}$ (inset of Fig. 5a). The pH of separate solutions of **1** and **2** was then adjusted from 10 to 2.5 using HNO$_3$ solution.

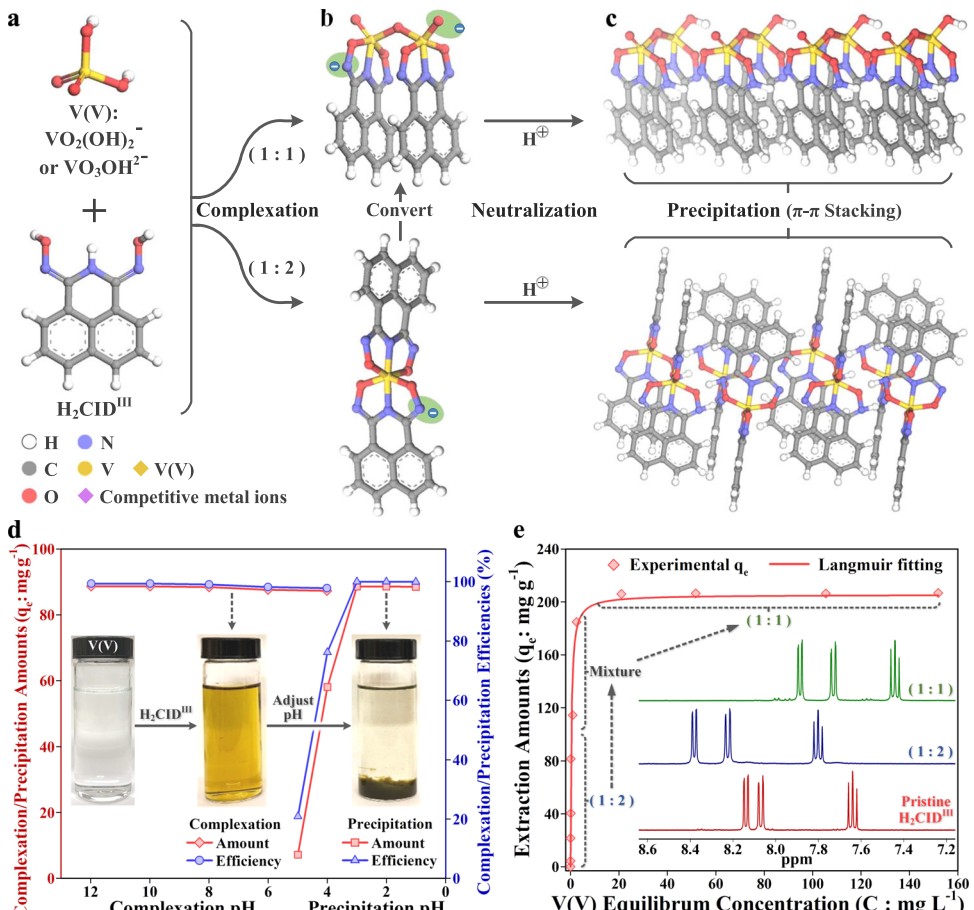

**Fig. 3 | Tandem complexation-precipitation technique for recovery of vanadium. a** Structures of V(V) and $H_2CID^{III}$. **b** 1:1 and 1:2 V-$CID^{III}$ complexes **1** and **2**. **c** neutralized and π-π stacked V-$CID^{III}$ complexes. V(V) recovery from single-metal aqueous solutions. **d** Influence of pH on V(V) complexation and precipitation; In each case, 0.05 g (0.25 g l$^{-1}$) of $H_2CID^{III}$ was exposed to 20 mg l$^{-1}$ V(V). **e** The extraction isotherm of V(V) on $H_2CID^{III}$; In each case, 0.05 g (0.25 g l$^{-1}$) of $H_2CID^{III}$ was exposed to V(V) for 12 h at 25 °C (Inset shows $^1$H NMR of pristine $H_2CID^{III}$, **2** and **1** obtained from samples in isotherm with $q_e$ of 80 mg g$^{-1}$ and 200 mg g$^{-1}$, respectively). Source data for (**d**) and (**e**) are provided as a Source Data file.

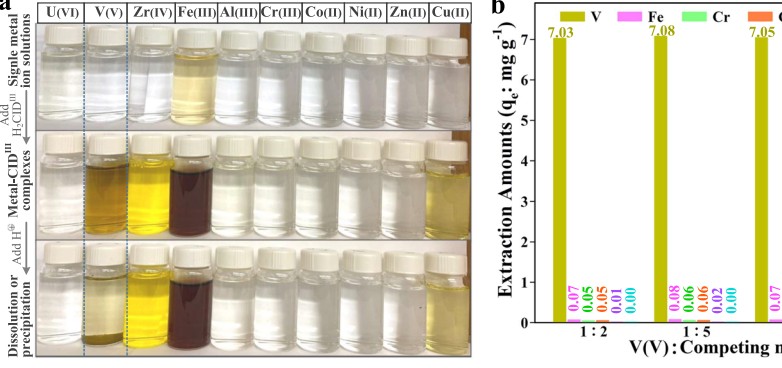

**Fig. 4 | V(V) recovery from mixed-metal aqueous solutions. a** Control experiment of $H_2CID^{III}$ in the recovery of several metal ions from aqueous solutions. Qualitatively, color change of solutions depict complexation. Precipitation of solely the V sample is observed. In each case, 0.01 g of $H_2CID^{III}$ was exposed to 1 mg l$^{-1}$ respective metal solutions for 12 h at 25 °C. **b** Quantitatively, extraction performances of $H_2CID^{III}$ for various metals from simulated mixed solutions containing V(V) and excess Fe(III), Cr(III), Cu(II), Ni(II), and Zn(II). Source data for (**b**) are provided as a Source Data file.

The titration curves of both **1** and **2** exhibited two stages of pH decrease, from 10.5 to 9.5 and from 7.0 to 5.8, for the complexation and neutralization processes, respectively (Fig. 5a). An abrupt pH change at roughly 4.0 indicated the formation of V·$CID^{III}$ precipitates. The pH ranges for complexation, neutralization, and precipitation extracted from titration curves agreed with the earlier results of pH influence on V(V) complexation and precipitation, validating the complexation of $H_2CID^{III}$ with $VO_3OH^{2-}$ or $VO_2(OH)_2^-$ across the investigated pH range (Fig. 3d).

FT-IR spectra of unchelated $H_2CID^{III}$, $Na_2$**1**, $Na_1$**2** and complexes obtained from the complexation and precipitation experiments (H$^+$-form) verified the protonation sites during neutralization. The bands at 3450 cm$^{-1}$, 3300 cm$^{-1}$, 1620 cm$^{-1}$ and 750 cm$^{-1}$ in pure $H_2CID^{III}$ can be attributed to the N−H stretching vibration in imide (−C−NH−C−), O−H

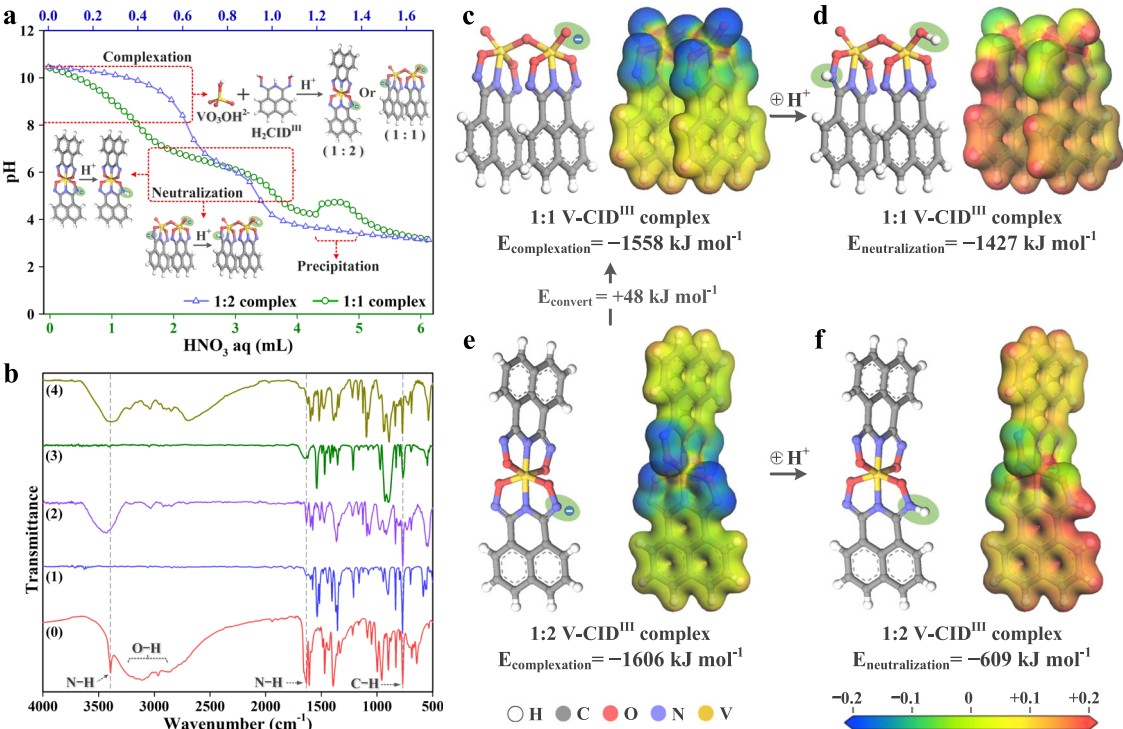

**Fig. 5 | Spectroscopic analyses and theoretical computations via DFT methods.** **a** Potentiometric titration curves of 1:2 and 1:1 V-CID$^{III}$ complexes. **b** FT-IR spectra of pristine H$_2$CID$^{III}$ (0), Na$^+$-form (1) and H$^+$-form (2) 1:2 V-CID$^{III}$ complexes, and Na$^+$-form (3) and H$^+$-form (4) 1:1 V-CID$^{III}$ complexes. **c** $E_{complexation}$ and electrostatic potential maps (EPM) of 1:1 V-CID$^{III}$ complex. **d** $E_{neutralization}$ and EPM of neutral 1:1 V-CID$^{III}$ complex. **e** $E_{complexation}$ and EPM of 1:2 V-CID$^{III}$ complex. **f** $E_{neutralization}$ and EPM of neutral 1:2 V-CID$^{III}$ complex. Source data for (**a**) are provided as a Source Data file.

stretching vibration in the oxime (−C=N−OH) group, N−H bending in imide and oxime groups, and naphthyl C−H bending, respectively (Fig. 5b, spectrum (0))[49]. Based on the differences in these peaks after acidification, the N in the oxime group (−C=N−O) is the only site in **2** that can accept a proton during neutralization, whereas in **1**, both the N in the oxime group and the O in the V−O group can be protonated (Supplementary Discussion 3). The acid stability (at pH 1.5) and complexation of different metal complexes were studied by $^1$H NMR spectra (Supplementary Fig. 23). Following a 1-h exposure, only the V(V) retained coordinated CID$^{III}$. In contrast, all other metals lacked coordination with H$_2$CID$^{III}$, leaving pure CID$^{III}$ ligands as residues. These results indicated the high acid resistance of the V-CID$^{III}$ complex compared with a range of relevant competing metal ions. The acid stability of the V-CID$^{III}$ complex can be attributed to the high charge density of V(V) and the basicity of H$_2$CID$^{III}$ donor atoms, enabling robust interaction and complexation even at low pH (pH 1.5). The acid resistance of V-CID$^{III}$ is highlighted, it is also essential to note that the key factor for selectivity lies in the anionic nature of V-CID complexes, with acid resistance playing a practical role in implementation.

The high V(V) affinity, the anionic nature and high acid resistance of V-CID$^{III}$ complexes are all key to enabling selective complexation and precipitation, as depicted in Fig. 3a–c. We further sought to understand the importance of the aromatic (naphthyl) moiety on H$_2$CID$^{III}$ so the performance of H$_2$CID$^{I}$ (i.e.: with no aromatic resonance stabilization) for complexation and precipitation of V(V) was investigated. The results showed that while complexation of V(V) occurred, there was no precipitation at low pH (Supplementary Fig. 24), presumably owing to the lack of π-π stacking to drive precipitation.

## Theoretical computations

To gain further insights into the complexation of V(V) and Fe(III) with CID$^{III}$ and the neutralization behaviors of these metal-CID$^{III}$ complexes, we computed the complexation energies ($E_{complexation}$) and neutralization energies ($E_{neutralization}$) of **1**, **2**, and **3** using DFT methods (Fig. 5c–f). As depicted in Fig. 5e, c, the $E_{complexation}$ of **2**, is −1606 kJ mol$^{-1}$, slightly higher than that of **1** (−1558 kJ mol$^{-1}$), suggesting a slightly higher preference for the formation of **2**. This finding aligns with the calculated bond lengths presented in Fig. 2. Nevertheless, due to the small energy gap, **2** can readily convert to **1** in the presence of excess V(V) in solution, as observed in the isotherm with a $q_e$ value around 200 mg g$^{-1}$ (Fig. 3e). As demonstrated in Fig. 5d, f, the $E_{neutralization}$ of **1** and **2** are −1427 and −609 kJ mol$^{-1}$, respectively. This implies that both **1** and **2** can be easily neutralized and form more stable complexes at low pH conditions. In contrast, for **3** (Supplementary Fig. 25a, b), the $E_{complexation}$ of the deprotonated complex is positive (+751 kJ mol$^{-1}$ in Supplementary Fig. 25b), while the $E_{complexation}$ of the neutralized complex is negative (−603 kJ mol$^{-1}$ in Supplementary Fig. 25a). This indicates that the 1:1 Fe-CID$^{III}$ complex is only stable in its protonated state, which is consistent with the observed neutral (protic) 1:1 Fe-CID$^{III}$ complex by crystallography (Supplementary Fig. 10). Furthermore, the $E_{complexation}$ of the V-CID$^{III}$ complexes is significantly higher than that of the neutral Fe-CID$^{III}$ complex, confirming the high stabilities of the V-CID$^{III}$ complexes.

Electrostatic potential maps of **1** and **2** were obtained based on the DFT calculations (Fig. 5c, e). The maps illustrate significant negative (blue isosurfaces) electrostatic potential on the N in the oxime group (−C=N−O) and O in the V−O group, indicating their propensity to accept protons during neutralization. The charges of oxygen and nitrogen in the V-CID$^{III}$ complexes and Fe-complex were also recorded (Supplementary Fig. 26). In **1**, the negative charge on N in the oxime group (0.63) is comparable to that of a terminal O in a V−O group (0.67). This confirms that both N in oxime groups (−C=N−O) and O in V−O groups in **1** can bind protons during neutralization. Conversely, in **2**, the negative charges of N in oxime groups (0.65) are significantly higher than those of O in V−O groups (≤0.30). Consequently, only the oxime N in **2** tends to protonate during neutralization, corroborating the FT-IR spectra (Fig. 5b).

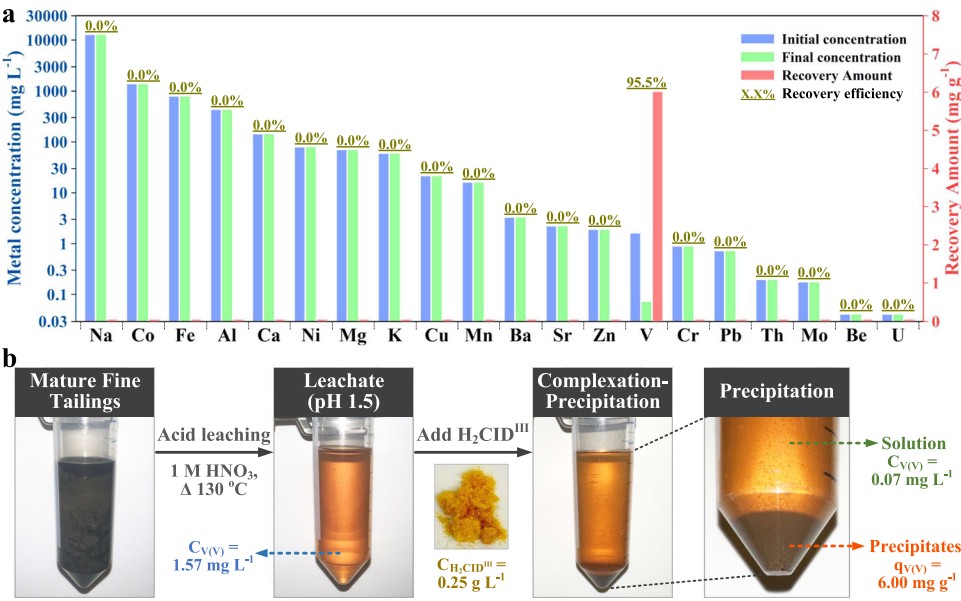

**Fig. 6 | V(V) extraction from real waste source. a** Selective recovery of V(V) with $H_2CID^{III}$ from oil sands tailings containing a series of metal ions. **b** Images of samples utilized and obtained during the extraction process. Source data are provided as a Source Data file.

## Vanadium extraction from oil sands tailings

A real-world sample of industrial wastewater offers a significantly greater challenge than any synthetic mixture of metal ions. To show practical applicability, $H_2CID^{III}$ was employed to recover V(V) from processed oil sand tailing waste obtained through a source-anonymized test bank operated by a government agency (see the "Methods" section). The leachate of oil sand tailing waste contained 1.57 mg l⁻¹ of V(V), along with various competing metal ions, some in high excess (Supplementary Table 9). The total organic carbon content in the leachate was 695 mg l⁻¹. The leachate was adjusted to pH 1.5 and treated with 0.05 g (0.25 g l⁻¹) of $H_2CID^{III}$ under room temperature and stirring conditions. Under this acidic condition, the complexation and precipitation process occur simultaneously. After a 30-min exposure period, $H_2CID^{III}$ demonstrated a V(V) recovery of 6.0 mg g⁻¹ with a recovery efficiency of 95.5%, resulting in a residual V(V) concentration of 0.07 mg l⁻¹. Interestingly, if the test was stopped after 5 min, $H_2CID^{III}$ could achieve a 90% recovery efficiency. In contrast, the recovery of competing metal ions by $H_2CID^{III}$ was negligible (Fig. 6a). These outcomes suggest the capability of $H_2CID^{III}$ to selectively bind and precipitate V(V) (Fig. 6b) in the presence of excess competing metal ions and dissolved organics, making it a promising candidate for the practical extraction of V(V) from waste sources.

Given its advantages over alternative materials (Supplementary Table 10), $H_2CID^{III}$ emerges as the ideal candidate for bulk extraction of V(V), owing to its ability to physically separate V(V) without a support, eliminating the complexities and limitations of using traditional heterogeneous systems. This independence leads to the benefits of using $H_2CID^{III}$ in terms of simplicity, and efficiency, primarily due to the absence of mass transfer limitations in supported V(V) extraction. The results from the extraction process, including its high V(V) capacity and selectivity at ambient conditions, set a benchmark for other materials. This system and process, coupled with a scalable synthesis method, show $H_2CID^{III}$'s advantages: cost-effectiveness, enhanced performance, and practical application suitability.

## Ecotoxicity assessment−zebrafish assay

The toxicity evaluation focused on the vanadium complexes **1** and **2**, which exhibited increased solubility in aqueous solutions. Zebrafish embryos were exposed to different concentrations of the compounds (ranging from 1 to 20 mg l⁻¹) for 72 h (see the "Methods" section).

Throughout this exposure period, the mortality and hatching rates at 24-, 48-, and 72-h post-fertilization (hpf) were monitored closely (Supplementary Fig. 27). Across all tested concentrations, there were no indications of any harmful effects or toxicity. Furthermore, no phenotypic abnormalities were observed after an extended observation period of up to 120 hpf (Supplementary Fig. 28). These findings suggest that the vanadium complexes do not exhibit toxicity to zebrafish at the tested concentrations. As a result, there is confidence in their safety for potential applications.

## Discussion

In this work, we present an improved approach for synthesizing $H_2CID^{III}$, an acid-stable chelating agent tailored for V(V) recovery. $H_2CID^{III}$ offers the distinct advantage of enabling pristine-material-based V(V) extraction without the need for material attachment to a support, a common requirement in other extractants resulting in a heterogeneous extraction process hampered by slow reactant mass transfer. Scalability of $H_2CID^{III}$ synthesis further enhances its practical viability. Notably, $H_2CID^{III}$ forms strong and distinctive complexes with V(V), leading to anionic 1:1 oxido, **1** and 1:2 non-oxido, **2** complexes, enabling selective precipitation in the presence of competing metal cations. Its advantageous V(V) complexation features include excellent acid resistance, rapid adsorption kinetics, high adsorption capacities over a broad pH range, and easy desorption and reusability. Method development simplifies the ligand-based extraction from stepwise complexation then precipitation to a one-step process applicable with oil sand waste sources, validating the results obtained from simulated studies. Furthermore, the toxicity potentials of V-CID^{III} complexes **1** and **2** investigated through zebrafish toxicity assessment assays suggest the complexes are non-toxic. These findings hold promise for addressing challenges in vanadium extraction, including V(V) depletion in natural sources and mitigating environmental and health-related concerns associated with extraction processes.

## Methods
### Synthesis of $H_2CID^{III}$

According to standard procedures outlined in literature[48,50], $H_2CID^{III}$ was synthesized by reacting the corresponding nitriles with 35% hydroxylamine in a methanolic solution (Supplementary Fig. 3). The improved synthesis of $H_2CID^{III}$ involved a 2-step process starting with

the reaction of 1,8-dibromonaphthalene (97%, Oakwood) (1.0 g, 3.49 mmol) and copper cyanide (99%, Aldrich) (2.7 g, 30 mmol) in 50 ml DMF at 130 °C for 12 h. After cooling to room temperature, 90% of DMF was distilled and the reaction mixture was poured into water, resulting in a precipitate. The precipitate was filtered, treated with aqueous ammonium hydroxide solutions (30% NH₃) and extracted with ethyl acetate (0.52 g, 84% yield). In the second step, H₂CID^III was obtained by reacting 1,8-naphthalenedicarbonitrile (0.5 g, 2.83 mmol) and 35% hydroxylamine (97%, Oakwood) (0.86 ml, 14.1 mmol) in 40 ml methanol. The reaction mixture was stirred at 80 °C for 12 h, and the desired product was obtained after evaporation of the solvent (0.6 g, 94% yield).

## Synthesis of crystals of V-CID^III complexes
For the synthesis of the crystals of 1:1 and 1:2 V-CID^III complexes, stoichiometric amounts of sodium metavanadate and H₂CID^III were reacted in an H₂O/MeOH solution for 5 h at room temperature. The dark brown solutions obtained were freeze-dried, and the solids were redissolved in acetone. Shiny black crystals of 1:1 and 1:2 V-CID^III complexes were obtained by vapor diffusion of ether into the acetone solution. To obtain the 1:1 Fe-CID^III crystals, a similar synthetic route was employed, using ferric chloride as the metal source and an excess of pyridine. Additionally, yellow crystals of H₂CID^III were grown through slow evaporation from methanol.

## Characterization of materials
All NMR data were acquired on Bruker Avance III 400 and 600 MHz spectrometers. ESI-MS was used to determine the mass of ligands and monitor the complexation of H₂CID^III with V(V). The experiments were conducted on an Agilent 6520 Q-TOF mass spectrometer. Aliquots of the methanolic solutions with ligand and the V-CID^III complexes were injected into the instrument and sprayed in the negative ion mode at 1 µl min⁻¹. The experimental conditions were set at 200 °C, 7 l/min drying gas flow, and the nebulizer was 12 psig. The concentrations of V(V) and interfering metals were measured by an ICP-OES, Varian 725-ES. The crystals obtained in this study were selected and mounted on a glass loop using Paratone. Diffraction experiments were performed on a Bruker Smart diffractometer equipped with an Incoatec Microfocus (graphite monochromated Cu Kα, $\lambda = 1.54178$ Å) and an APEX II CCD detector. The crystal was kept at 173 K during data collection. Diffraction spots were integrated and scaled with SAINT[51], and the space group was determined with XPREP[51]. Using Olex2[52], the structure was solved with the ShelXT[53] structure solution program using Intrinsic Phasing and refined with the ShelXL[54] refinement package using Least Squares minimization. Potentiometric analyses were completed using an Orion 960 Titrator (Thermo Electron Corporation). A 10 ml aliquot of each sample was titrated with 0.01 M nitric acid until a pH of 2.5 was reached. End-point determination was accomplished via double differentiation of titration results. Functional group detection analysis was performed on a Nicolet 4700 FTIR spectrometer.

## V(V) extraction experiments
In experiments to determine the pH effect, extraction capacity, kinetics and desorption, complexation and precipitation were performed in steps. The effect of pH on complexation and precipitation of V(V) were determined by treating a 200 ml 20 mg l⁻¹ V(V) spiked deionized water with 0.05 g (0.25 g l⁻¹) H₂CID^III for 12 h at room temperature. The pH range for complexation and precipitation were kept at 4–12 and 1–5, respectively, and adjusted with HNO₃ and NaOH solutions. The extraction isotherm experiments were conducted using a 200 ml V(V) spiked deionized water with concentrations ranging from 5 to 200 mg l⁻¹ and treated with 0.05 g (0.25 g l⁻¹) of H₂CID^III for 12 h at room temperature. The extraction kinetics experiments were conducted with initial V(V) concentrations of 20 and 50 mg l⁻¹ and treated with 0.05 g (0.25 g l⁻¹) of H₂CID^III for specified duration at room

temperature. The pH of complexation and precipitation was adjusted and kept at 8 and 2, respectively. All solutions from complexation and precipitation were filtered through a 0.2 micrometer syringe filter and were analyzed by ICP-OES to determine residual metal concentrations. The complexation and precipitation amount of V(V) were calculated by using:

$$q = \frac{\left(C_i - C_f\right) \times V}{m} \quad (1)$$

where $q$ (mg g⁻¹) refers to the V(V) extraction amount; $C_i$ and $C_t$ (mg l⁻¹) represent the initial and final V(V) concentration, respectively; $V$ (l) refers to the volume of solution, and $m$ (mg) is the mass of the used H₂CID^III ligand.

Thiourea was employed as a desorbing agent, leveraging its established capability to liberate firmly bound metal ions from adsorbents, in the quest to release V(V) and regenerate H₂CID^III. The precipitate recovered after the extraction process was subjected to treatment with hydrochloric acid (HCl) solutions at concentrations of 1 M and 5 M. Simultaneously, acidified thiourea solutions containing 0.1 M HCl and ranging from 0.2 M to 2 M in thiourea concentration were utilized, all subjected to stirring for a duration of 1 h. The decomplexation efficiency (DE) was calculated using the following equation:

$$DE = \frac{\text{mass of V(V) in the desorption solution (mg)}}{\text{mass of } V(V) \text{ complexed with CID}^{III} \text{(mg)}} \times 100\% \quad (2)$$

## Metal ion competition assay
To test the selectivity of H₂CID^III to V(V) in the presence of competing metal ions, the concentrations of V(V) to Fe(III), Cr(III), Cu(II), Ni(II) and Zn(II) were kept at ratios 1:1, 1:2, 1:5, 1:10 and 1:20. Selectivity was assessed in varying pH, time, and ligand amount conditions. In this experiment, the acidity of the solutions was adjusted by HNO₃ solution. Each metal extraction capacity was calculated as described earlier and compared to ascertain selectivity. Hereafter, the distribution coefficient ($K_d$) was calculated using:

$$K_d = \frac{q}{C_f} \quad (3)$$

And the selectivity coefficient ($\beta$) was calculated as follows[55]:

$$\beta_{V/\text{competing metal}} = \frac{k_V}{k_{\text{competing metal}}} = \frac{q_V}{q_{\text{competing metal}}} \times \frac{C_{\text{competing metal}}}{C_V} \quad (4)$$

## Computational methods
To investigate the complexation energies ($E_{\text{complexation}}$) of V(V) and Fe(III) with CID^III, and the neutralization energies ($E_{\text{neutralization}}$) of these metal-CID^III complexes, density functional theory (DFT) calculations were performed using DMol3 tools in Material Studio (Accelrys Software Inc.). In the DFT methods, local density approximation (LDA) with the PWC functions, double numeric polarization (DNP) basis set, and DFT semi-core Pseudopots approximation were utilized to compute the energies involved in "Density of states", "Electron density", "Electrostatics", "Fukui function", "Orbitals" and "Population analysis". The cleaved structures taken from crystal structures of 1:1 V-CID^III complex, 1:2 V-CID^III complex and 1:1 Fe-CID^III complex (Fig. 2 and Supplementary Fig. 10) were used to simulate the local environments of complexation and neutralization. The value of the complexation and neutralization energies ($E_{\text{complexation}}$ and $E_{\text{neutralization}}$) were calculated as the energy difference before and after complexation or neutralization,

respectively, as defined by:

$$E_{complexation} = E_{Metal-CID} - (E_{Metal} + E_{CID}) \tag{5}$$

$$E_{neutralization} = E_{H-[Metal-CID]} - (E_{Metal-CID} + E_H) \tag{6}$$

where $E_{Metal-CID}$ is the energy of 1:1 V-CID$^{III}$, 1:2 V-CID$^{III}$ and 1:1 Fe-CID$^{III}$ complexes, while $E_{Metal}$, $E_{CID}$ and $E_H$ are the energies of V(V) ion, Fe(III) ion, CID$^{III}$ and H$^+$. $E_{H-[Metal-CID]}$ is the energy of metal complexes after protonation[56]. In addition, DMol3 electrostatic potential maps of 1:1 V-CID$^{III}$, 1:2 V-CID$^{III}$ and 1:1 Fe-CID$^{III}$ complexes were obtained based on DFT calculations[57].

## V(V) extraction from oil sand tailings

Oil sand tailings waste, known as Mature Fine Tailings (MFT), generated from oil sands extraction via steam-assisted methods in Alberta, Canada, were obtained from InnoTech Alberta. The viscous oil-laden mixture was dried at 120 °C for 24 h, followed by pulverization. Subsequently, it was subjected to nitric acid leaching at 130 °C and used for V(V) extraction studies under the previously established optimum conditions. The detailed metal ion concentrations of the leached solution are shown in Supplementary Table 9.

## Ecotoxicity assessment−zebrafish assay

The toxicity assessment was performed following the procedures discussed by Osin et al.[58]. AB wild-type adult zebrafish (Danio rerio) were maintained under controlled conditions in a fish breeding circulatory system at a temperature of $28 \pm 0.5$ °C and a light/dark cycle of 14 h:10 h. The zebrafish were fed live brine shrimps (Artemia salina) twice daily. Prior to spawning, two pairs of male and female fish were placed in a mating box with a divider separating them. After 1 day, the divider was removed in the morning to trigger spawning, and the embryos were collected 2 h later. Healthy and fertilized embryos at 4 h post-fertilization (hpf) were selected under a stereomicroscope (Olympus-SZ61, Olympus Ltd., Japan) and placed in U-bottom 96-well plates (Costar-3599, Corning, US) with one embryo per well.

Notably, H$_2$CID$^{III}$ displayed negligible solubility in water but became soluble in the presence of V(V), ensuring convenient recovery post-desorption. Consequently, the toxicity assessment prioritized the vanadium complexes; **1** and **2** due to their increased solubility in aqueous solutions. Respective wells were filled with 200 μl of varying concentrations of **1** and **2**, along with E3 medium as negative controls. Two replicates were performed for each treatment, each consisting of 16 embryos. Statistical analyses were conducted using Graphpad Prism version 9 for windows, Graphpad software (Graphpad, Boston, Massachusetts USA). Differences between the control and vanadium complex treated groups were analyzed using two-way ANOVA followed by Tukey's multiple comparisons test. The $p$ value threshold of 0.05 ($p < 0.05$) was used to define statistical significance. The developmental progress of the zebrafish embryos was observed at 24, 48, and 72 hpf. The toxicological endpoints evaluated included hatching interference, phenotypic abnormalities, and mortality (necrosis of the embryos). All experiments were conducted in compliance with the protocols approved by the Animal Ethics Committee at Tongji University, with approval granted by the Animal Center of Tongji University (Protocol #TJAD-004-22B02). The data obtained were presented as average values.

## Reporting summary

Further information on research design is available in the Nature Portfolio Reporting Summary linked to this article.

## Data availability

The authors declare that all the data supporting this study's findings are available within the article (and Supplementary Information files) or can be obtained from the corresponding author upon request. The X-ray crystallographic coordinates for the structures reported in this study have been deposited at the Cambridge Crystallographic Data Center (CCDC) with deposition numbers 2299690-2299693. These data can be obtained free of charge from the Cambridge Crystallographic Data Center via https://www.ccdc.cam.ac.uk. Source data are provided with this paper.

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

## Acknowledgements

We thank the Natural Sciences and Engineering Research Council (NSERC) of Canada for support of this work. O.A.O. is grateful for a Vanier Canada Graduate Scholarship and an Izaak Walton Killam Doctoral Scholarship. S.L.J.L. and S.J.L. express gratitude to the National Natural Science Foundation of China #22176150. The authors thank Mr. Michael Nightingale in the Geosciences Department at the University of Calgary for his support in the ICP-OES measurements and potentiometric titration analyses. Lastly, some of the data from this manuscript was previously published in O.A.O.'s thesis (hdl.handle.net/1880/118023).

## Author contributions

G.K.H.S. conceived, fundraised and supervised the project. O.A.O. designed and planned the project, performed syntheses, extraction experiments, and data analysis. Shuo L. performed DFT calculations, contributed to data analysis, and modified figures. B.S.G. performed X-Ray crystallographic analysis on the complexes reported. S.L.J.L. and S.J.L. performed the zebrafish toxicity assessment. O.A.O., Shuo L. and G.K.H.S. contributed to the project discussions and manuscript drafting. All authors approved the final version.

## Competing interests

We have filed a provisional patent on this approach with O.A.O. and G.K.H.S. as inventors. The remaining authors declare no competing interests.
