## [Peer Review File · Nature Communications]

A molecular extraction process for vanadium based on tandem selective complexation and precipitationREVIEWER COMMENTS

Reviewer #1 (Remarks to the Author):

The authors have conducted a study focusing on the extraction of vanadium ions utilizing naphthalimidedioxime. The investigation yields intriguing findings pertaining to the synthesis of naphthalimidedioxime and its efficacy in selectively extracting vanadium ions. Nonetheless, I have a few comments that I recommend addressing prior to the publication of this manuscript.

Comment

1. Methods should be more detailed provided.

(a) The procedure for measuring the yield of H₂CIDIII during synthesis using a novel method

(b) The procedure for obtaining the extraction isotherm shown in Fig. 3 (e), considering that the detailed procedure is anticipated to differ from that used in the vanadium (V) extraction experiments outlined in the Methods on page 19.

(c)

2. Does "complexation pH" refer to the initial pH or the equilibrium pH? This clarification is crucial, as the pH is expected to undergo changes during complexation and precipitation, particularly in acidic conditions, as inferred from Fig. 5.

3. Regarding the extraction mechanism, it is presented that the high acid resistance of V-CIDIII is key to enabling selective complexation and precipitation. However, I believe the expression used may not be accurate. The experimental result in Supplementary Figure 22 reflects a singular behavior and does not constitute a pivotal aspect of the reaction mechanism. While those results are important, understanding the reason for the high acid resistance of V-CIDIII is crucial for enabling selective complexation and precipitation.

4. Is the complexation reaction of VO₃OH²⁻ with H₂CIDIII general for all pH regions? The dominant species of V(V) ions varies with pH, so the complexation reaction may vary accordingly, and kinetics may also be significantly affected by pH.

5. Have you ever studied the extraction behavior of Mo or W ions, particularly those presenting as anions in alkaline regions, similar to V ions using H₂CIDIII? This suggestion could provide valuable hints for examining the extraction mechanism of ions using H₂CIDIII.

6. Additional review on the recovery of V should be incorporated into the Introduction section. Many studies have reported the effective recovery of V from waste sources.

7. This demonstrates an excellent effect in terms of selectivity. Therefore, it may prove effective in removing trace amounts of vanadium, from soil-polluted water. However, the adsorption capacity is notably low compared to previously reported ion adsorption resins, suggesting a potential limitation in its application for the vanadium recovery process from resources. Further discussion on this aspect is requested.

Reviewer #2 (Remarks to the Author):

This paper optimized the synthesis method of H₂CIDIII. Detailed discussion was conducted on the structure, complexation mechanism, kinetics, and other aspects of the complex. Industrial applications were simulated using oil sands tailings. This method has strong innovation and application value.

1. The authors used metal chelating agent H₂CIDIII to recover vanadium. In the introduction section, other types of metal chelating agents should be compared to highlight the advantages of the chelating agents used by the authors.
2. Was the crystal information of the crystals formed by H₂CIDIII and vanadium calculated using DFT? Suggest supplementing the crystal with XRD characterization.
3. When the pH was around 4, the precipitation efficiency was very low, and in other pH ranges, the precipitation efficiency was high. The authors need to explain this phenomenon in a mechanistic way.
4. The chelating agent had a very high selectivity for competitive metal cations. A brief explanation of the separation mechanism is needed.
5. To compute tendency to protonate, it is essential to calculate Fukui indices. Also, is the selected calculation method (DMol3 tools in Material Studio) suitable for the system?
6. How to recover unreacted H₂CIDIII dissolved in the aqueous phase?
7. It is necessary to emphasize and characterize the toxicity of H₂CIDIII.

Response to Reviewers

We are thankful to the reviewers for their valuable and insightful comments, which have enabled us to improve the quality of the manuscript. All comments/suggestions are responded to below. Moreover, we have carefully revised the manuscript to avoid any language, technical and scientific errors. The revised text in the manuscript was marked with the **red color** as a “for review” version.

Manuscript No.: NCOMMS-23-52300-T

Title: A molecular extraction process for vanadium based on tandem selective complexation and precipitation

Authors: Oluwatomiwa A. Osin, Shuo Lin, Benjamin S. Gelfand, Stephanie Ling Jie Lee, Sijie Lin, George K. H. Shimizu*

Reviewer #1 (Remarks to the Author):

The authors have conducted a study focusing on the extraction of vanadium ions utilizing naphthalimidedioxime. The investigation yields intriguing findings pertaining to the synthesis of naphthalimidedioxime and its efficacy in selectively extracting vanadium ions. Nonetheless, I have a few comments that I recommend addressing prior to the publication of this manuscript.

Comments

1. Methods should be more detailed provided.

(a) The procedure for measuring the yield of $\text{H}_2\text{CID}^{\text{III}}$ during synthesis using a novel method.

Response:

Thank you for the comment. The equation for calculating the yield of $\text{H}_2\text{CID}^{\text{III}}$ in our synthesis method has been included in the SI and emphasized in red color. The yield (Y) can be calculated using the following formula:

$$Y (\%) = \frac{\text{the experimentally obtained quantity of } \text{H}_2\text{CID}^{\text{III}}}{\text{the maximum possible quantity of } \text{H}_2\text{CID}^{\text{III}}} \times 100$$

The experimentally obtained quantity of $\text{H}_2\text{CID}^{\text{III}}$ was obtained by weighing, following the purification of the final product. On the other hand, the maximum possible quantity of $\text{H}_2\text{CID}^{\text{III}}$ represents the theoretical yield, calculated based on stoichiometry. This calculation relies on the balanced chemical equation for the reaction:

The excess reagent in the reaction is CuCN, and the yield was calculated based on limiting reagent, ArBr_2 .

(b) The procedure for obtaining the extraction isotherm shown in Fig. 3 (e), considering that the detailed procedure is anticipated to differ from that used in the vanadium (V) extraction experiments outlined in the Methods on page 19.

Response:

The following details have been added, emphasized in red color, in the V(V) extraction experiments section in “Methods”.

In our experimental approach for the V(V) extraction isotherm, 200 mL of V(V) spiked deionized water with concentrations ranging from 5 to 200 mg L⁻¹ was treated with 0.05 g (0.25 g L⁻¹) of H₂CID^{III} for 12 hours at room temperature. Figure 3e illustrates the isotherm generated through our complexation-precipitation process. Specifically, for this experiment, the complexation was maintained at pH 8, and the precipitation was set at pH 2 to ensure that all of the formed V(V)-CID complexes crashed out of solution. After the equilibration periods for both complexation and precipitation, the concentration of the metal in the liquid phase was quantified via ICP-OES analysis. The extraction capacity of V(V) was calculated using the following formula:

$$q = (C_i - C_f) V / m$$

where q (mg g⁻¹) refers to the V(V) extraction amount; C_i and C_f (mg L⁻¹) represent the initial and final V(V) concentration, respectively; V (L) refers to the volume of solution, and m (mg) is the mass of the used H₂CID^{III} ligand.

2. Does "complexation pH" refer to the initial pH or the equilibrium pH? This clarification is crucial, as the pH is expected to undergo changes during complexation and precipitation, particularly in acidic conditions, as inferred from Fig. 5.

Response:

In our study, the reference to complexation pH pertains specifically to the equilibrium pH in the synthetic solution. However, for the wastewater experiments, we did not adjust the pH during the experiment. The initial pH was 1.5, and significant changes in pH were not anticipated after the extraction process. This point has been added to the text.

3. Regarding the extraction mechanism, it is presented that the high acid resistance of V-CID^{III} is key to enabling selective complexation and precipitation. However, I believe the expression used may not be accurate. The experimental result in Supplementary Figure 22 reflects a singular behavior and does not constitute a pivotal aspect of the reaction mechanism. While those results are important, understanding the reason for the high acid resistance of V-CID^{III} is crucial for enabling selective complexation and precipitation.

Response:

Thank you for your insightful comment. Your concerns are valid, and we acknowledge the need for clarity in describing the pivotal aspects of the reaction mechanism.

We would like to provide a more accurate representation of the acid resistance of V-CID^{III} and its role in enabling selective complexation and precipitation. The acid stability of the

vanadium complexes can be attributed to the high charge density of V(V) and the basicity of the donor atoms on H₂CID^{III}. The strong interaction strength between V(V) and H₂CID^{III} gave rise to the formation of a rare non-oxido 1:2 pentavalent vanadium complex, 2. Typically, V(V)-organic ligand complexes tend to maintain oxido bonds in the VO₂⁺ moiety, leading to the creation of 1:1 complexes due to steric hindrance and limited coordination sites. Consequently, pentavalent non-oxido complexes are rare.

Like our investigation, Leggett et al. also observed that the complexation of V(V) with two H₂CID^I ligands resulted in the formation of a distinctive non-oxido complex.¹ Consequently, the stability constant of the V-CID^I complex, with a log β value of 53.5, was determined to be the highest among V(V) complexes.²

Consequently, in this study, the strong interaction between vanadium and H₂CID^{III} allows for complexation even at low pH values (pH 1.5), as evidenced in the selectivity assessment studies (Supplementary Fig. 20). Interestingly, the acid stability of the resulting complexes ensures the survival of the vanadium complexes under harsh acidic conditions.

The NMR experiment presented in Supplementary Figure 22 was designed to illustrate the weak interaction between H₂CID^{III} and other metal ions at such low pH, emphasizing that all other metal ions either do not react or are eluted under these conditions. While the acid resistance of V-CID^{III} is highlighted, it is essential to note that the key factor for selectivity lies in the anionic nature of V-CID complexes, with acid resistance playing a supplementary role. This point has been noted in red color in the 'vanadium extraction mechanism' section.

Reference: 1. Leggett, C. J.; Parker, B. F.; Teat, S. J.; Zhang, Z.; Dau, P. D.; Lukens, W. W.; Peterson, S. M.; Cardenas, A. J. P.; Warner, M. G.; Gibson, J. K.; Arnold, J.; Rao, L. Structural and Spectroscopic Studies of a Rare Non-Oxido V(v) Complex Crystallized from Aqueous Solution. *Chem. Sci.* 2016, 7 (4), 2775–2786.

2. Ivanov, A. S.; Leggett, C. J.; Parker, B. F.; Zhang, Z.; Arnold, J.; Dai, S.; Abney, C. W.; Bryantsev, V. S.; Rao, L. Origin of the Unusually Strong and Selective Binding of Vanadium by Polyamidoximes in Seawater. *Nat. Commun.* 2017, 8 (1), 1–9.

4. Is the complexation reaction of VO₃OH₂⁻ with H₂CID^{III} general for all pH regions? The dominant species of V(V) ions varies with pH, so the complexation reaction may vary accordingly, and kinetics may also be significantly affected by pH.

Response:

Thank you for your thoughtful comment. As demonstrated in Supplementary Figure 22, our investigation has accounted for the variation in the dominant species of V(V) ions. Specifically, it shows that the prevalent species of V(V) are VO₃OH₂⁻ or VO₂(OH)₂⁻ during complexation across the pH range of 5 to 12. Consequently, our experimental design has carefully considered the variation in V(V) species with respect to pH and based on the results presented in Figure 3d (pH-dependent extraction studies), we can state that the complexation with VO₃OH₂⁻ or VO₂(OH)₂⁻ shows no significant difference across the examined pH range. The interaction between V(V) and H₂CID^{III} remained robust and is not significantly influenced by pH. The high strength of interaction between V(V) and H₂CID^{III}, as illustrated in our results, suggests that the complexation reaction is generally consistent across different pH regions.

Similarly, $\text{H}_2\text{CID}^{\text{III}}$ demonstrated effective binding to V(V), even in acidic conditions where the prevailing species is VO_2^+ (refer to Supplementary Fig. 22). This observation is supported by the results of isotherm and kinetics studies conducted during the one-step extraction process (see Supplementary Fig. 18 and 19).

Based on our experimental findings, the interaction between V(V) and $\text{H}_2\text{CID}^{\text{III}}$ appears to be stable and not sensitive to variations in pH. While we acknowledge the variability in dominant species of V(V) ions with pH, our study focuses on the overall strength and stability of the V-CID^{III} interaction, which remains a key aspect of our investigation.

5. Have you ever studied the extraction behavior of Mo or W ions, particularly those presenting as anions in alkaline regions, similar to V ions using $\text{H}_2\text{CID}^{\text{III}}$? This suggestion could provide valuable hints for examining the extraction mechanism of ions using $\text{H}_2\text{CID}^{\text{III}}$.

Response:

In response to your inquiry, we would like to provide clarification regarding the scope and focus of our study, which centers on oil sands tailings. The selection of competing metal ions (Fe(III), Cr(III), Cu(II), Ni(II) and Zn(II)) in this investigation was guided by the relative concentration of each metal ion in the tailings, as detailed in Supplementary Table 9.

Our primary focus was on metals ions with concentrations similar to or higher than that of vanadium in the tailings. However, molybdenum (Mo) exhibited a concentration an order of magnitude lower than vanadium, and tungsten (W) ions were not present in the waste. Nonetheless, the extraction studies from oil sands tailings revealed that $\text{H}_2\text{CID}^{\text{III}}$ exhibited no preference for Mo, and thus, Mo was not extracted from the waste. Consequently, it can be inferred that the interaction strength between Mo and $\text{H}_2\text{CID}^{\text{III}}$ was insufficient to withstand the harsh conditions of the waste source, or the charge of Mo-CID^{III} may be either cationic or neutral.

We hope that this explanation offers clarity regarding the specific objectives and research scope of our study.

6. Additional review on the recovery of V should be incorporated into the Introduction section. Many studies have reported the effective recovery of V from waste sources.

Response:

Thank you for your feedback. We agree that this is an important aspect of our research, and we have included a review of relevant studies in the introduction to enhance the overall context of our work. These details are also mentioned in our response to question 7 and emphasized in red color in the introduction section of the main text.

7. This demonstrates an excellent effect in terms of selectivity. Therefore, it may prove effective in removing trace amounts of vanadium, from soil-polluted water. However, the adsorption capacity is notably low compared to previously reported ion adsorption resins, suggesting a potential limitation in its application for the vanadium recovery process from resources. Further discussion on this aspect is requested.

Response:

Thank you for the comment. Indeed, recent literature highlights the efficacy of diverse materials, including porous sorbents, ionic liquids, nanoparticles, and modified composites in Vanadium extraction from aqueous solutions, with impressive capacities ranging from 240 to 712.4 mg/g. However, a comprehensive evaluation of material selectivity for V(V) in the presence of competing metal ions (Fe(III), Cr(III), Cu(II), Ni(II), Zn(II), Cl⁻, etc.) is lacking and could impact these capacities. As a result, the material's selectivity for the target species becomes crucial for accurately evaluating its performance in the extraction process.

These points have been included in the introductory section of the main text.

Reviewer #2 (Remarks to the Author):

This paper optimized the synthesis method of H₂CID^{III}. Detailed discussion was conducted on the structure, complexation mechanism, kinetics, and other aspects of the complex. Industrial applications were simulated using oil sands tailings. This method has strong innovation and application value.

1. The authors used metal chelating agent H₂CID^{III} to recover vanadium. In the introduction section, other types of metal chelating agents should be compared to highlight the advantages of the chelating agents used by the authors.

Response:

Thank you for your thoughtful feedback. We appreciate your suggestion to provide a comparison of different types of metal chelating agents in the Introduction section, emphasizing the advantages of the specific chelating agent (H₂CID^{III}) employed in our study. We agree that highlighting the distinctive features and benefits of our chosen chelating agent in comparison to others will contribute to a more comprehensive understanding of our research. We revised the introductory section (marked in red) accordingly to address this point and provide a more thorough analysis of the rationale behind selecting H₂CID^{III} for Vanadium recovery.

2. Was the crystal information of the crystals formed by H₂CID^{III} and vanadium calculated using DFT? Suggest supplementing the crystal with XRD characterization.

As shown in the Methods section, the synthesis of the crystals of 1:1 and 1:2 V-CID^{III} complexes involved reacting stoichiometric amounts of sodium metavanadate and H₂CID^{III} in an H₂O/MeOH solution for 5 hours at room temperature. The dark brown solutions obtained were freeze-dried, and the solids were redissolved in acetone. Shiny black crystals of 1:1 and 1:2 V-CID^{III} complexes were obtained by vapor diffusion of ether into the acetone solution. A

similar synthetic route using ferric chloride as the metal source was adopted to obtain the 1:1 Fe-CID^{III} crystals. Additionally, yellow crystals of H₂CID^{III} were grown through slow evaporation from methanol. Hereafter, the structures of these crystals were resolved through single crystal X-ray diffraction (SC-XRD) and Rietveld refinement (Fig. 2, Supplementary Tables 2 and 3 and XRD patterns shown in Supplementary Fig. 9). This point has been mentioned and emphasized in red color in the ‘Synthesis and characterization of H₂CID^{III} and vanadium complexes’ section within the Results and Discussion.

Supplementary Figure 9 | XRD patterns of ligand and complexes obtained from single crystal

3. When the pH was around 4, the precipitation efficiency was very low, and in other pH ranges, the precipitation efficiency was high. The authors need to explain this phenomenon in a mechanistic way.

Response:

We appreciate the opportunity to delve into the details and provide an understanding of this phenomenon.

In the acidic environment (pH 3 or below), there is an abundance of protons in the solution. The increased proton concentration allows for the effective neutralization of anionic vanadium complexes present in the system. The enhanced coordination between V(V) ions and H₂CID^{III}, coupled with the effective neutralization of anionic vanadium complexes, contributes to a substantial increase in precipitation efficiency. The formation of stable, neutral complexes allows for efficient and rapid precipitation of vanadium species from the solution. This phenomenon is experimentally addressed via potentiometric titration and results are shown in Fig. 5a.

However, around pH 4, the protonation of the anionic complexes may be less pronounced, leading to a reduced ability to effectively neutralize anionic vanadium complexes. This results in a lower precipitation efficiency compared to more acidic conditions.

4. The chelating agent had a very high selectivity for competitive metal cations. A brief explanation of the separation mechanism is needed.

Response:

We appreciate your attention to the selectivity of the chelating agent in our study. Nevertheless, there is some uncertainty about the specific chelating agent mentioned—whether it is H_2CID^{III} or thiourea. Nevertheless, we provide explanations for both scenarios.

Firstly, in the case of H_2CID^{III} , it exhibited notable selectivity towards V(V) in the presence of other metal ions at similar or higher concentrations. The selectivity mechanism is tandem involving both complexation and precipitation based on the charge of the resulting complexes; specifically, V- CID^{III} forms anionic complexes, while other metal- CID^{III} complexes are either cationic or neutral. This difference in charge allows for effective separation via precipitation through pH adjustment. The detailed explanation of this selectivity mechanism is provided in the 'selectivity mechanism' section of the manuscript.

In addition, considering thiourea as a complexing agent, the separation mechanism relies on its ability to efficiently elute strongly bound metal ions from adsorbents. This capability arises from the competitive nature of thiourea for metal ions, facilitated by synergistic effects from electrostatic interactions, covalent binding of lone pairs on the nitrogen atom, and the presence of multiple coordination sites (S and N).

We hope this clarifies the separation mechanisms associated with both H_2CID^{III} and thiourea in our study.

5. To compute tendency to protonate, it is essential to calculate Fukui indices. Also, is the selected calculation method (DMol3 tools in Material Studio) suitable for the system?

Response:

Thank you for the comment. The DMol3 tools are well-suited for our system, as detailed in the "Methods" part. In our DFT calculations, we employed the local density approximation (LDA) with PWC functions, a double numeric polarization (DNP) basis set, and the DFT semi-core Pseudopotentials approximation to compute energies, ensuring enhanced precision. The calculated energies include "Density of states," "Electron density," "Electrostatics," "Fukui function," "Orbitals," and "Population analysis," significantly broadening the scope of our calculations.

Additionally, Nucleophilic Attack (Fukui(+)) for both 1:1 and 1:2 complexes was determined through DFT calculations. As indicated in the subsequent table, the differences in Fukui(+) indices for N and O in the 1:1 and 1:2 complexes are insignificant. However, the charges of N and O in the V-complexes vividly depict the electrostatic potential of each N and O atom in the complexes.

1:1 complex		1:2 complex		Charges of N and O in V-complexes	
Atom	Fukui(+)	Atom	Fukui(+)		
V(1)	0.032	V(1)	0.061		
V(2)	0.033	O(2)	0.036		
O(3)	0.034	O(3)	0.041		
O(4)	0.034	O(4)	0.039		
O(5)	0.036	O(5)	0.039		
O(6)	0.033	N(6)	0.043		
O(7)	0.042	N(7)	0.016		
O(8)	0.019	N(8)	0.041		
O(9)	0.042	N(9)	0.040		
N(10)	0.039	N(10)	0.016		
N(11)	0.014	N(11)	0.040		
N(12)	0.045				
N(13)	0.044				
N(14)	0.015				
N(15)	0.041				

6. How to recover unreacted H₂CID^{III} dissolved in the aqueous phase?

Response:

H₂CID^{III} exhibits inherent insolubility in water. Consequently, upon desorption, V(V) is separated from CID and concentrated within the solution. Meanwhile, any unreacted CID and regenerated CID persist as solids in the solution, allowing for their retrieval through filtration and centrifugation.

In the revised manuscript, we articulate this information to enhance the comprehension of the recovery process for unreacted H₂CID^{III}.

7. It is necessary to emphasize and characterize the toxicity of H₂CID^{III}.

Response:

Thank you for raising the important concern regarding the toxicity of H₂CID^{III}.

To address this issue, we conducted an ecotoxicity assessment using zebrafish. However, we want to emphasize that H₂CID^{III} displayed negligible solubility in water but became soluble in the presence of V(V), ensuring convenient recovery post-desorption. Consequently, the toxicity assessment prioritized the vanadium complexes; 1 and 2 due to their increased solubility in aqueous solutions.

To assess the toxicity potentials of **1** and **2**, zebrafish embryos were exposed to different concentrations of the compounds (ranging from 1 to 20 mg/L) for 72 hours. Throughout this exposure period, the mortality and hatching rates at 24-, 48-, and 72-hours post-fertilization (hpf) were monitored closely. Across all tested concentrations, there were no indications of any harmful effects or toxicity. Furthermore, no phenotypic malfunctions were observed after an extended observation period of up to 120 hpf. These findings suggest that the vanadium complexes do not exhibit toxicity to zebrafish at the tested concentrations. As a result, there is confidence in their safety for potential applications.

The new data has been added to the supplementary information and highlighted in red color

Supplementary Figure 27 | Toxicity assessment. **a and b**, Mortality rate (4%). **c and d**, Hatching rate (~100%) of zebrafish embryos treated with **1** and **2**.

Embryos exposed to samples

Supplementary Figure 28 | Representative microscopic images of zebrafish embryos exposed to 1 and 2 obtained at 48 and 120 hpf

REVIEWERS' COMMENTS

Reviewer #1 (Remarks to the Author):

The manuscript has undergone thorough revision and is now considered suitable for publication in this journal. However, one suggestion/comment is still pending.

1. "This point has been noted in red color in the 'vanadium extraction mechanism' section." was suggested in the response letter as reply to ask of No. 4.. However, it could not be found. I recommend you to add the explanation in the response letter into the manuscript

Reviewer #2 (Remarks to the Author):

The author gives very detailed answers to the questions raised earlier. Articles can be published.

REVIEWERS' COMMENTS

Reviewer #1 (Remarks to the Author):

The manuscript has undergone thorough revision and is now considered suitable for publication in this journal. However, one suggestion/comment is still pending.

1. "This point has been noted in red color in the 'vanadium extraction mechanism' section." was suggested in the response letter as reply to ask of No. 4.. However, it could not be found.

I recommend you to add the explanation in the response letter into the manuscript

Response: Thank you for your feedback. The manuscript now incorporates an explanation for the robust acid resistance of V-CID^{III} and elucidates the primary mechanism for achieving selectivity.

Reviewer #2 (Remarks to the Author):

The author gives very detailed answers to the questions raised earlier. Articles can be published.

Response: We would like to express our appreciation for the author's dedication and thorough review of our manuscript.